# Response-Surface-Methodology-Based Increasing of the Isotropic Thermal Conductivity of Polyethylene Composites Containing Multiple Fillers

**DOI:** 10.3390/polym15010039

**Published:** 2022-12-22

**Authors:** Hannelore Ohnmacht, Rudinei Fiorio, Tom Wieme, Dagmar R. D’hooge, Ludwig Cardon, Mariya Edeleva

**Affiliations:** 1Centre for Polymer and Material Technologies (CPMT), Department of Materials, Textiles and Chemical Engineering, Ghent University, Technologiepark, 130, Zwijnaarde 9052, 9000 Ghent, Belgium; 2Department of Circular Chemical Engineering, Faculty of Science and Engineering, Maastricht University, P.O. Box 616, 6200 MD Maastricht, The Netherlands; 3Centre for Textiles Science and Engineering (CTSE), Department of Materials, Textiles and Chemical Engineering, Ghent University, Technologiepark, 70A, Zwijnaarde 9052, 9000 Ghent, Belgium; 4Laboratory for Chemical Technology (LCT), Department of Materials, Textiles and Chemical Engineering, Ghent University, Technologiepark, 125, Zwijnaarde 9052, 9000 Ghent, Belgium

**Keywords:** thermal conductivity, carbon nanotubes, response surface methodology, hybrid filler composite, polymer composites, aluminum oxide, graphite

## Abstract

To optimize the thermal conductivity of high-density polyethylene, 15 hybrid filler composites containing either aluminum oxide, graphite, expanded graphite, carbon nanotubes or a combination of the former, have been studied using an extrusion-compression processing tandem. The experimental density of the cube-shaped specimens is substantially lower than the theoretical density calculated by the linear mixing rule, mainly for the composites with high filler contents. The morphology of the composites, as studied by scanning electron microscopy (SEM), highlighted a good dispersion quality and random orientation of the fillers in the test specimens but also revealed air inclusions in the composites, explaining the density results. It is shown that the addition of filler(s) increases both the melt viscosity (up to ca. 270%) and the thermal conductivity (up to ca. 1000%). Hence, a very strong increase of TC can be practically hampered by a too high viscosity to enable processing. Supported by ANOVA analysis, the application of response surface methodology (RSM), assuming a perfect compression, indicates that all fillers have a significant effect on the thermal conductivity and synergistic effects can be achieved. The regression model obtained can adequately predict the thermal conductivity of composites of various compositions, as already confirmed based on three validation experiments in the present work.

## 1. Introduction

The studying of polymer composites presenting a high thermal conductivity (TC) has been gradually increasing in the recent years [1,2]. These high-thermal conductive polymer-based composites can play an important role in commercial and industrial thermal management applications, such as heat sinks and heat transfer devices [3,4]. They show many advantages compared to metallic counterparts, such as good corrosion resistance, low density, low cost, and easy processability [2]. 

Unfortunately, inherent defects in bulk polymer materials such as impurities, voids, and unregulated entanglements, in combination with a random orientation of polymer chains, result in small phonon mean free pathways, thus, low TC [2,5]. As most polymers tend to display a low intrinsic TC in the range of 0.1–0.5 W.m^−1^.K^−1^ [6], they are often combined with fillers that are inherently possessing high TC values to increase the ability to conduct or dissipate heat. These fillers can vary in material, type, and size [7,8,9,10]. However, in order to obtain significantly high TC values, high filler loadings are often necessary, usually leading to brittleness, high melt viscosity, and limited processability [11]. Aiming to avoid these drawbacks, the combination of fillers in various types and shapes is common practice for thermoplastic composites for TC applications. Upon proper design, a continuous filler network with enhanced filler connectivity results, even using lower filler contents [2] and complementary with recent developments for piezoresistive sensors [12]. In addition, a more efficient packing can be acquired, and the viscosity can be decreased [13]. Since the packing density can be optimized using a hybrid filler system, synergistic effects can be evoked, resulting in a higher TC compared to the case of an isolated filler [14,15,16,17]. It has been specifically observed that carbon nanotubes (CNTs) in combination with other fillers can generate a three-dimensional thermal transfer pathway between the primary fillers as a result of their high aspect ratio [18,19]. The secondary fillers, being the CNTs, promote the interconnection between the other fillers so that heat conduction through the material is facilitated [20].

Alongside experimental research theoretical routes have been explored to design conductive polymer composites. Notably to predict the overall TC in thermoplastic composites, various theoretical models have already been developed to estimate the properties of possible candidate materials and to limit the number of experiments. A difference can be made between analytical (less generally applicable) micro-mechanical models and (more detailed) finite element methods (FEM) [2,21,22,23,24,25]. FEM can provide fairly accurate estimations of the composite TC as these simulations consider the composite morphology and interfacial resistance between matrix and filler [25,26]. FEM is, however, very time-consuming and many model parameters for both the filler and matrix material need to be known. This explains why synergistic effects for hybrid filler systems cannot be studied with FEM as the model is still too simplified [2]. Practically, one thus needs to rely on the simpler models but limited theoretical work exists for more complex hybrid systems.

The aim of the present work is, therefore, to set up an analytical model able to predict the hybrid-filler-based composite TC, in which synergistic effects among different fillers can be accounted for. Three fillers of varying shapes and sizes are considered, i.e., aluminum oxide (Al_2_O_3_; sphere-shaped), graphite (G; platelet-shape) and expanded graphite (EG; platelet-shape). A high-density polyethylene (HDPE) matrix, always doped with 1.0 m% CNT, is used as reference. Response surface methodology (RSM) is applied to evaluate the effects of the different formulations, especially on the composite TC.

## 2. Materials and Methods

### 2.1. Materials

Extrusion-grade high-density polyethylene (HDPE, Lupolen 4261AG, Lyondellbasel, Rotterdam, The Netherlands) was used as matrix material for the composites. Three fillers were varied in their content, i.e., aluminum oxide (Al_2_O_3_, DAM 45, Denka, Tokyo, Japan—spherical-shaped fillers) with an average particle size D50 of 41.2 μm; Graphite (G, KS44, Timrex, Imerys, Paris, France) with a D50 of 15.8 μm; and expanded graphite (EG, GFG75, Sigratherm, SGL Carbon, Wiesbaden, Germany) with a D50 of 75 μm, the latter two being platelet-shaped fillers. In addition, carbon nanotubes (CNTs) with an average length of 1.5 μm in the form of a thermoplastic masterbatch (15.0 m%) with HDPE as matrix (Plasticyl HDPE1501, Nanocyl, Sambreville, Belgium) were added to include a total, fixed content of 1.0 m% CNTs into each composite.

### 2.2. Experimental Methods

#### 2.2.1. Processing Methods from Raw Materials to Final Isotropic Sample

Figure 1 presents the processing steps that were used to produce the composite samples. Firstly, the polymer and fillers were micro-compounded in a twin-screw extruder. The extrudate was then manually cut into pellets of ±5 mm. After that, these pellets were compression-moulded, generating cube-shaped test samples. It should be noted that processing can affect the orientation of non-spherical fillers in the material. During extrusion, these fillers are likely oriented in the flow direction due to shear so that composite pellets may have anisotropic properties [27]. However, during compression moulding, the pellets were deliberately placed in the mould in a random configuration and were subjected to heat and pressure so that the overall orientation of the fillers in the bricks can be considered random. Therefore, this study assumes isotropic properties for the polymer composites. In what follows, the compounding and compression moulding are discussed in more detail.

For the compounding of the samples, a Thermo Scientific HAAKE Minilab twin-screw extruder was used, as shown in Figure 1a. The bypass channel that is integrated in the set-up allows for the recirculation of material in order to obtain a better homogenization of the compounded materials. The loaded volume of the polymer and fillers in the mini-extruder was 7 to 7.5 cm³. Every batch was recirculated for 5 min, while the screw speed was increased with 5 rpm every minute from 10 to 30 rpm for optimal mixing. This procedure was repeated 5 times for every compound in order to obtain sufficient material for compression moulding. The (average) processing temperature was 220 °C and the extrudate was cooled in open air at room temperature and then manually pelletized.

The composite pellets were compression moulded into cubes of ±8 cm³ at a temperature of 200 °C. The material was first heated in the mould (Figure 1c) for 10 min, allowing the pellets to melt. After that, the material was compressed for 5 min under a pressure of ca. 50 kN. The cube was then removed from the mould, the material completely cooled down and solidified. For every compound, two samples were compression moulded.

#### 2.2.2. Characterization Methods

The isotropic thermal conductivity of the compression moulded samples was determined using the transient plane source method (Hot Disk TPS 2500S, Hot Disk AB, Göteborg, Sweden) according to ISO 22007-2. Each sample was measured at least 3 times using the Kapton 5465 sensor. The measurement time and heat output were varied between 5 and 20 s, and 120 and 150 W, respectively. The probing depth of the measurements was varied between 3 and 4 mm.

Density measurements were performed in air and ethanol (96%) according to the ISO1183-1, method A, using a precision analytical balance (Precisa XR 205SM-DR, Precisa Gravimetrics AG, Dietikon, Switzerland). For every compound, the density is given as the calculated average of two compression moulded samples.

Scanning electron microscopy (SEM) images were taken using a Phenom PRO Desktop SEM (Phenom-World, Eindhoven, The Netherlands). The three fillers (Al_2_O_3_, G, and EG) were dispersed on an adhesive carbon or aluminum tape and analyzed. For the composites, thin slices (2–3 mm) were cut from the moulded bricks with a saw, and then cryogenically fractured after immersion in liquid nitrogen. The surface of this brittle fracture was investigated by SEM to verify the dispersion of the fillers and overall morphology. 

By measuring the extrudate mass output ($Q_{m}$; kg/s) and calculating the melt density (kg/m³) of the compositions, the volumetric flow rate ($Q_{v}$; m³/s) can be determined. In turn, the apparent (wall) shear rate (${\dot{\gamma }}_{a}$) and wall shear stress (τ) can be calculated using [28]:(1)$${\dot{\gamma }}_{a}=\frac{6\times {\;Q}_{v}}{w\;\times \;h²}$$
(2)$$\tau =\frac{h\times w\times \Delta P}{2\times \left(h+w\right)\times \Delta L}$$
in which h, w and ΔL refer to the height, width, and length of the slit die (m) and ΔP is the pressure drop (Pa) along the capillary length ΔL. This pressure drop can be obtained from the two pressure transducers that are integrated in the slit-capillary backflow channel of the micro-compounder, as shown in Figure 1a.

The melt viscosity of the compounds was measured according to a common experimental procedure to obtain basic rheological properties of non-Newtonian fluids using a micro rheology compounder [28]. By plotting the logarithm of the apparent viscosity ($\eta_{a}$) as function of the logarithm of the apparent shear rate, the power law index or shear thinning factor (n) can be obtained from the slope of the linear regression (n=slope+1). The true (wall) shear rate (${\dot{\gamma }}_{true}$) can then be calculated using n via the modified Rabinowitsch relationship:(3)$${\dot{\gamma }}_{true}=\left(\frac{2+n\;}{3}\right)\times {\dot{\gamma }}_{a}$$

The viscosity (η) can then be determined as the ratio of the τ and $\dot{\gamma }$:(4)$$\eta =\frac{\tau }{\dot{\gamma }}$$

### 2.3. Theoretical Framework of Response Surface Methodology

To study the effect of more than two (statistical) variables, factorial designs are interesting [29]. In such a design, all possible combinations of so-called factors are investigated in each trial. In this research, it was chosen to study three factors, namely the fillers Al_2_O_3_, G and EG, each at two levels, with level being the concentration of filler in the composite. In other words, a 2³ factorial was chosen. The content of CNTs remained always fixed at 1.0 m%. 

All compositions of the factorial design can be geometrically presented in a cube. Figure 2 shows the geometric view and Table 1 presents the design matrix of the face-centered central composite design (FCCCD), with the orange points in Figure 2 representing the factorial points, the blue points being the face-centered points, and the green point being the center point of the cube. In this case, 0 m% was chosen as low level (coded as −1) for the three factors (Al_2_O_3_, G, and EG), 45.0 m% was chosen as high level (coded as +1) for the Al_2_O_3_, while for both graphites, G and EG, 12.5 m% was chosen as the high level. The reason for the difference between the high levels for Al_2_O_3_ and G and EG can be found in the shape of the fillers. As Al_2_O_3_ is spherical and has, therefore, isotropic thermal properties, a higher value as +1 is chosen. G and EG are platelet-shaped fillers and have anisotropic thermal properties but are used to improve the conductive path as it has been stated that a combination of different filler shapes can lead to a synergistic improvement in the thermal conductivity. When only spherical fillers are used, higher filler loadings are generally needed to obtain a high thermal conductivity [1]. In addition, G and EG are used for their commercial relevance, as these fillers are relatively cheap and effective to increase the TC. Hereby, G is cheaper than EG but both are added to investigate possible differences in TC increasing effect. For all high levels of the factors, the composite consists of a maximum of 71 m% filler, with 1.0 m% from CNTs, so that there is still a polymeric fraction of 29 m% to guarantee the processability. 

The center point, which is indicated as (0, 0, 0) or the green dot in Figure 2, was replicated three times in the design (Run 9a, 9b and 9c) to allow an independent estimation of errors. The six-star points were added in Figure 2 in order to have enough data points to be able to calculate a second-order response equation. These star points are situated in the middle of every plane of the cube to obtain a face-centered central composite design. The results were analyzed by the JMP software, considering a significance level of 0.05.

The design matrix in Table 1 holds all 15 compositions that were used to obtain the response surface model. The sequence of compounding, moulding, and analysis of the samples were randomly conducted to independently distribute nuisance factors (errors). The regression equations were obtained considering coded variables $X_{1}$, $X_{2}$ and $X_{3}$ representing Al_2_O_3_, G and EG, respectively, and with −1 ≤ $X_{i}$ ≤ 1 (coded levels for each factor). The mass content of each variable is coded according to Equation (5):(5)$$X_{i}=\frac{m_{i}-\frac{\left(m_{i_{low}}+m_{i_{high}}\right)}{2}}{\frac{\left(m_{i_{high}}-m_{i_{low}}\right)}{2}}$$
in which $X_{i}$ is the coded variable of the factor i (i= 1, 2, or 3), $m_{i}$ is the natural (uncoded) mass content of the factor i, $m_{i_{low}}$ is the natural lowest mass content of the factor i, and $m_{i_{high}}$ is the natural highest mass content of factor i.

## 3. Results and Discussion

### 3.1. Construction of the Response Surface Model

Table 2 shows the TC results of the compression moulded samples for all compositions from Table 1 in relative format, which is sufficient in the scope of the present work highlighting the relevance of making composites for enhanced TC. Run 9, the central point, is chosen as a reference (X_ref_ = 1) from which the thermal conductivity of the other samples is compared to. As these samples are assumed to have isotropic characteristics, the TC is assumed the same in all directions inside the cubes. From these results, it can clearly be seen that TC increases significantly with an increase in the amount of filler. Sample 8, which is the composition with the maximum amounts of filler used, reaches a TC well above 1 W.m^−1^.K^−1^, which is significantly higher than the TC of HDPE with 1.0 m.% CNTs (Sample 1).

The TC responses from Table 2 were analyzed using JMP statistical software with a significance level (α) of 0.05. For this analysis, main effects, interaction terms and quadratic effects of the different fillers were investigated for a face-centered central composite design (FCCCD) [29]. Subsequently, a second-order regression model could be obtained from these results to predict TC:(6)$$\begin{array}{ll} TC=1.4017+ & 0.7081\;X_{1}+0.5797\;X_{2}+0.7334\;X_{3}+0.2857\;X_{1}X_{2}\\ & +0.3376\;X_{1}X_{3}+0.1667\;X_{2}X_{3}+0.2157\;X_{1}^{2}+0.1187\;X_{3}^{2} \end{array}$$
in which the coded variables $X_{1}$, $X_{2}$ and $X_{3}$ represent Al_2_O_3_, G, and EG, respectively, with −1 ≤ $X_{i}$ ≤ 1 the coded levels for each factor. The determination coefficient (*R*²) of this regression is 0.997 and ANOVA results are included in the Appendix A. It follows that Al_2_O_3_ and the EG have the most major effects on the TC as a result of the high coefficients for these factors. For Al_2_O_3_, this is likely because the amount of fillers added to the system is much larger than for the other fillers; up to 45 m.% compared to, e.g., up to 12.5 m.% for the graphites. Furthermore, EG has a higher efficiency in increasing TC than G. As discussed in detail later on, the EG flakes used are much bigger than the G flakes, denoting that fewer polymer-filler interfaces are introduced to the system. Since these interfaces act as barriers to heat flow, they have a negative effect on the thermal conductive behaviour of the composite [8]. With EG flakes, there are less barriers and thus TC increases. 

From the analysis, it is also noticeable that the coefficient of the interaction terms and quadratic effects are also quite high and should not be neglected during the TC prediction. The quadratic effect of G ($X_{2}^{2}$) is not included in the equation, since this term is insignificant according to the statistical analysis.

Upon making a response matrix with the results from the regression model, the response surface plane can be plotted. Three surface planes are shown in Figure 3, with the results of the addition of three m% of Al_2_O_3_ also shown in in Figure 3a–c, respectively. The planes show a significant curvature, as a result of the synergistic effects of the interaction terms and quadratic effect among the different fillers themselves. Figure 3 also includes the experimental data points from Table 2.

### 3.2. Validation of the Response Surface Model

To validate the response surface model, three extra composites have been manufactured following the procedure as presented in Table 3. The concentrations of the fillers in these extra samples are randomly chosen points according to the geometric view of the factorial design in Figure 2. Table 3 (top part) shows the filler concentrations (in m%) of these new samples. The isotropic TC was measured from these blocks and the results are presented in Table 3 (bottom part), in combination with the expected TC values calculated from the regression model. Analyzing the results, it can be seen in Table 3 that the measured values are in line with the expected values from the proposed model. Taking into account the standard deviation of the TC measurements and the error margin of the measuring instrument, the model gives an accurate TC prediction.

### 3.3. Characterization and Analysis of the Cube-Shaped Samples

As the filler homogeneity in the polymer matrix is an important factor to consider for TC measurements, SEM has been used to evaluate the dispersion quality of the fillers in the composite. A distinction is made between analysis of the fillers themselves and the actual composites. 

In Figure 4, SEM-images are shown regarding the different shapes of the three fillers used with a 1150× magnification. Figure 4a shows a SEM image of the Al_2_O_3_ spheres from which it can be seen that the powder has a wide particle size range. The powder contains both large particles with a diameter of 50 μm and small particles that have a diameter of 10 μm and less. It can also be seen that only few spheres are broken, leading to an irregular particle shape. On the surface of the Al_2_O_3_ spheres, small sphere-shaped features are present (satellites) which are an artifact of the production method [30,31]. Figure 4b,c show the graphite and expanded graphite particles, respectively. These graphites are stacked layers of graphene that differ in size and TC. EG is modified graphite with an interlayer space between the graphene, leading to a higher surface area (and particle count) than the G flakes [32].

SEM-images of Sample 12 (22.5 m.% Al_2_O_3_/6.25 m.% EG) and Sample 11 (45 m.% Al_2_O_3_/6.25 m.% G/6.25 m.% EG) in Table 1 are shown in Figure 5 using a magnification of 300×. Analyzing these SEM-images, it follows that the fillers are well-dispersed in the HDPE matrix, both for Sample 12 in which only Al_2_O_3_ and EG were used, and for Sample 11 in which all fillers were used. The white Al_2_O_3_ particles can be clearly distinguished in both figures. It is sometimes difficult to identify the G and EG particles in the composite, but a few flakes are indicated with a blue dotted (G) and green full (EG) circle. There are also quite some spherical indentations visible (indicated in dashed red). When the samples were cryogenically fractured, the Al_2_O_3_ particles did not break into two halves, but stayed intact, leaving indentations in the other half of the sample. These SEM-images, in combination with the SEM-images of the other specimens, are also shown in the Appendix A. From the SEM-images, it can thus be concluded that all fillers are well-dispersed and randomly oriented so that the prepared composites are assumed to have isotropic thermal properties.

It should be stressed that a better dispersion does not necessarily lead to a higher TC, since this means that also more matrix-filler interfaces are present in the composite [8,33]. More important is that an efficient thermal conductive path can be formed so that phonons can travel in an unhindered manner. In this context, in Figure 6, a schematic representation of the conductive path of a HDPE hybrid filler system is made in case a high amount of fillers used. The HDPE matrix and CNTs are not shown for two practical reasons. Firstly, the polymer is supposed to fill all gaps between the fillers (white space). Secondly, the CNTs are very small compared to the other fillers (1.5 μm in length and nanometer-scale in cross-section in contrast to spherical Al_2_O_3_ fillers with a mean diameter of 41.2 μm) so that they are invisible in the SEM images.

From Figure 6, it can be seen that by using fillers with different size and shape, small filler particles can fill the gaps between bigger fillers. A good interconnectivity can therefore be created between the fillers leading to a good conductive path (orange line) and optimal packing density. Moreover, the not-depicted CNTs serve as TC bridges to further improve the interconnection. 

It Is important to support the TC and SEM analysis via viscosity data, as illustrated in Table 4 for 15 compositions for an actual (wall) shear rate of 3 s^−1^. During compounding screw speeds have been varied from 10 to 30 rpm according to Equations (1)–(4) [28]. These results show that the actual (wall) shear rates vary between 1 and 7 s^−1^ if the screw speed increases from 10 rpm to 30 rpm.

It follows from Table 4, that for Sample 8, in which the maximum amounts of fillers are used, an increase in viscosity of ca. 270% is observed, compared to the viscosity of Sample 1, being a composition of only HDPE and 1.0 m.% CNTs. The results in Table 4 have been further analyzed using JMP statistical software with a significance level of 0.05, from which a viscosity regression equation (Equation (7)) could be obtained: (7)$$\eta =12769.2+3820.0\;X_{1}+937.8\;X_{2}+2665.0\;X_{3}+846.2\;X_{1}X_{2}+1480.6\;X_{1}X_{3}$$

From this regression equation, no significant quadratic effect is observed and *R*² is as high as 0.983. The ANOVA results are included in the Appendix A, and it follows from Equation (7) that the fillers clearly result in a viscosity increase of the composite. As these fillers do not melt and remain solid during processing, they increase the resistance to flow.

The coefficient terms of both Al_2_O_3_ and EG are much higher than the coefficient of G. In addition, the interaction terms $X_{1}X_{2}$ and $X_{2}X_{3}$ in Equation (7) are considerably high, which can be explained by Al_2_O_3_ particles preventing platelet graphite particles from orienting in the direction of the flow, giving more resistance to the polymer melt which can result in a viscosity increase.

To determine the dominant filler for the increase in the composite viscosity, the viscosity can be predicted using the regression Equation (7) and Equation (5) to calculate the mass percentage into coded level. When the viscosity is calculated using 10 m% Al_2_O_3_ and no other fillers, and this viscosity is compared to the viscosity when only 10 m% G is used and when only 10 m% EG is used, significant differences in the viscosity at 3 s^−1^ can be obtained. An addition of 10 m% Al_2_O_3_ results in a viscosity of 8337 Pa.s, whereas an addition of 10 m% EG leads to a viscosity of 9568 Pa.s. An addition of 10 m% G results in the lowest viscosity of 7820 Pa.s. From this, it can thus be stated that EG is the dominant filler for the viscosity increase. The table with these results are added in the Appendix A. 

In parallel, density measurements have been performed to see if the parts have been properly formed during compression moulding. The results of these measurements have been compared to the theoretical density value according to the linear mixing rule:(8)$$\rho_{c}=\left(1-\phi \right)\rho_{p}+\sum \phi_{f}\rho_{f}$$
in which $\rho_{c}$, $\rho_{p}\;$ and $\rho_{f}\;$ are the density of the composite, polymer matrix and filler, respectively, and *ϕ* is the volume fraction of filler. The densities that have been used for each material are included in the Appendix A. It follows that Al_2_O_3_ has a significant higher density than the other fillers, and that HPDE has the lowest density. This means that the more fillers that are added to the composite, the heavier the material becomes. Table 5 holds the density results from both the experimental samples and the calculated results according to the mixing rule.

In order to analyse a possible trend between the experimental density results and the density according to the mixture rule, Figure 7 shows the density measurements of the prepared composites as function of the density according to the mixing rule. 

Figure 7 shows that there is a substantial difference between the (relative) experimental values (green) and (perfect) theoretical values according to the linear mixture rule (orange line). The difference is the smallest if no extra fillers (apart from CNTs) have been added to the composite and increases notably if higher amounts of fillers are used. In case no Al_2_O_3_ is used in the compositions, the density is significantly lower but the difference between the theoretical and experimental values is smaller, as seen in Figure 7 for Sample 1, 3, 5, 7 and 10 in Table 1. The addition of more Al_2_O_3_ leads to higher densities but also larger differences between the theoretical and experimental values. This is likely by the increase in viscosity with high amounts of Al_2_O_3_ and graphites added.

It should be noted that compression moulding of pellets and powders with high amounts of fillers never leads to a packing density of 100%, due to mismatches in the particle shape and due to the increased viscosity [34]. The polymer material, however, should in theory be able to fill these gaps, decreasing the number of voids, especially under these filler loadings. The low densities can therefore not be fully caused by a poor packing density but must be more linked to the processing conditions. To check the size and amount of air voids present in the samples, slices of the cubes have been cut (±5 mm) and cryogenically broken to obtain a brittle fracture surface. The orange circles in Figure 8 indicate the air voids, from which it is clear that there are numerous air gaps present, explaining the lower density results in Figure 7. 

These voids are thus likely a consequence of the method of compression moulding. In the present work, compression moulding has been performed in two steps, as presented in Figure 9. Firstly, the material inside the mould has been heated for 10 min without pressure to completely melt. After that, pressure has been applied to the mould until the point material started to flow out. It is possible that a too high viscosity of the composites could insufficiently allow air to escape (be trapped) during the compression step, instead of pushing out material. The higher the filler loading, the more viscous the composite, and the more difficult the air can escape, resulting in a higher porosity. The high viscosity of the highly filled composites may thus induce to the formation of compression moulded test specimens presenting too much air inclusions, strongly reducing the density (cf. Figure 7).

An important caveat to be added is that the model in the present work is based on experimental thermal conductivity resulting from the compression moulded samples. Since air has a TC of only 0.025 W.m^−1^.K^−1^, the simulated TC values might be lower than what would be achievable if no air was present. 

## 4. Conclusions

A face-centered central composite design has been set up to predict the thermal conductivity (TC) of a hybrid filler composite, taking into account synergistic effects. HDPE as matrix material has been combined with Al_2_O_3_, graphite (G), and expanded graphite (EG). For each compound, 1 m.% of CNTs has been added to the system to improve the TC pathway. The isotropic thermal conductivity of 15 compression moulded samples has been measured with variable filler amounts and these experimental data points have been analyzed by response surface methodology (RSM). 

The RSM results indicate that all fillers have significant effects on the TC, both individually and due to interactions thus synergism; Al_2_O_3_ and EG even showed significant quadratic effects. The regression model obtained in this study can accurately predict the TC, which has been confirmed by three extra validation compounds. The TC could be increased to ca. 1000%, with a consequent increase in the melt viscosity of ca. 270%. It is further shown that Al_2_O_3_ and EG have the highest effect on both melt viscosity and thermal conductivity. 

The density of the composites shows lower values than those predicted by the rule of mixture, mainly for the highly filled samples. SEM images indicate a good dispersion of the fillers but also the presence of air bubbles in the moulded test specimens, which could be related to the method of compression moulding and the system high viscosity. 

It can be overall concluded that the development of HDPE-matrix composites containing hybrid fillers can dramatically increase the TC due to synergistic and quadratic effects, making them more efficient than single-filler composites. However, the melt viscosity is increased by the addition of fillers as well so that a balance between a very high TC and acceptable processability may be challenging.

## Figures and Tables

**Figure 1 polymers-15-00039-f001:**
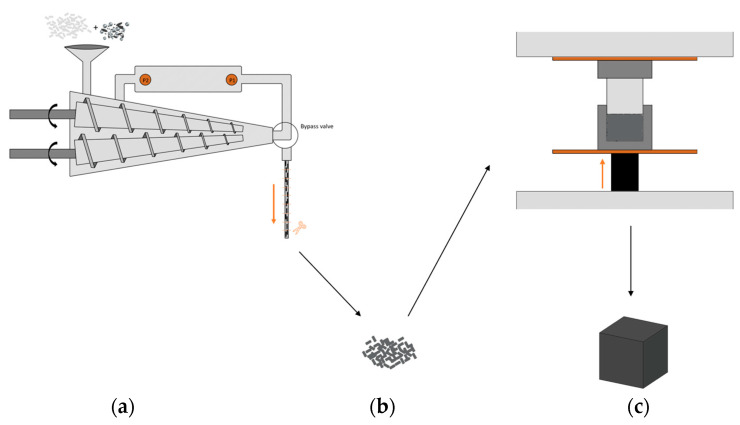
Overall processing scheme. (**a**) Twin-screw micro-compounder with pressure transducers (P1 and P2) integrated in a slit-capillary backflow channel. (**b**) Resulting composite pellets. (**c**) Compression moulding of these pellets in a cube-shaped sample.

**Figure 2 polymers-15-00039-f002:**
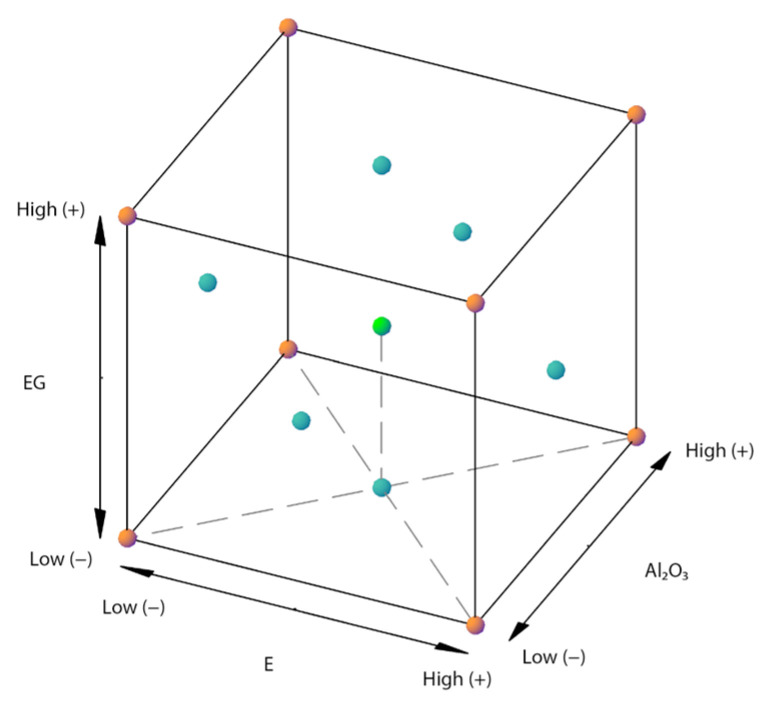
Geometric view of the FCCCD.

**Figure 3 polymers-15-00039-f003:**
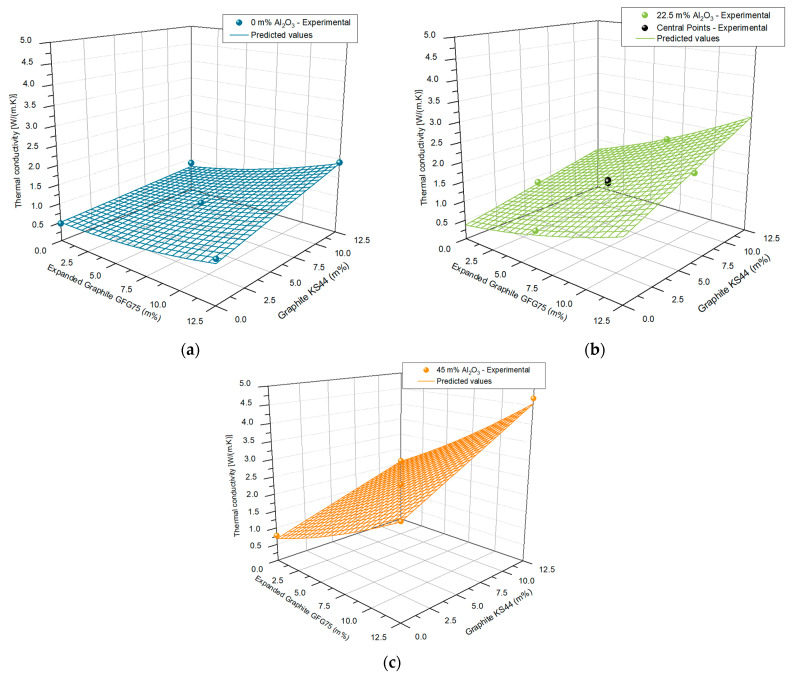
Response surface, in combination with the experimental data points for: (**a**) 0 m% Al_2_O_3_; (**b**) 22.5 m% Al_2_O_3_; (**c**) 45 m% Al_2_O_3,_ respectively.

**Figure 4 polymers-15-00039-f004:**
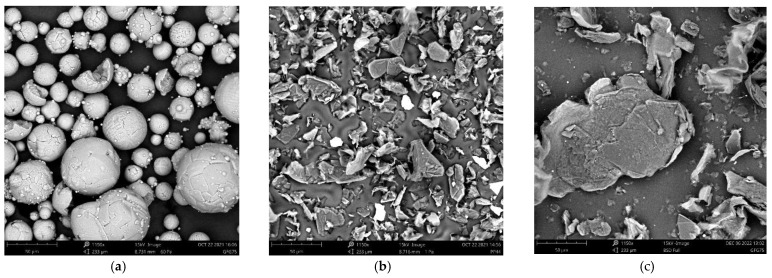
SEM images of the fillers: (**a**) Aluminum oxide (Al_2_O_3_); (**b**) Graphite (G); (**c**) Expanded Graphite (EG).

**Figure 5 polymers-15-00039-f005:**
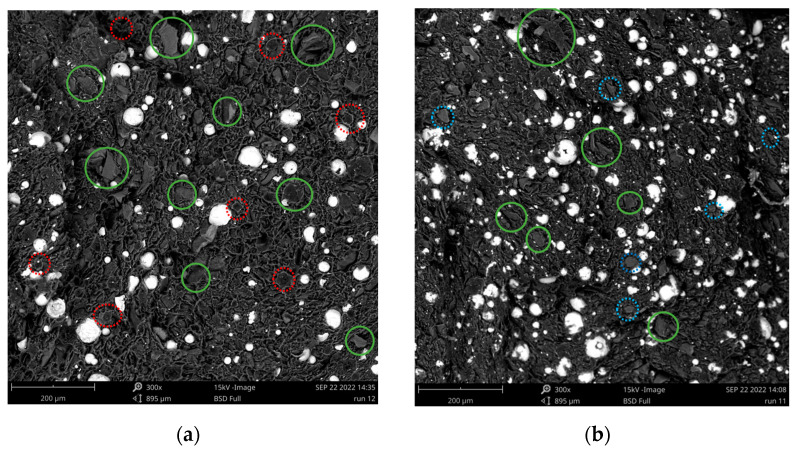
SEM images of the filler dispersion in the HDPE matrix; (**a**) Run 12 (22.5 m.% Al_2_O_3_/6.25 m.% EG); (**b**) Run 11 (45 m.% Al_2_O_3_/6.25 m.% G/6.25 m.% EG)—the green circles indicate the EG, the blue circles indicate the G and the red circles indicate spherical indentations. The white spheres in both figures are the Al_2_O_3_ particles.

**Figure 6 polymers-15-00039-f006:**
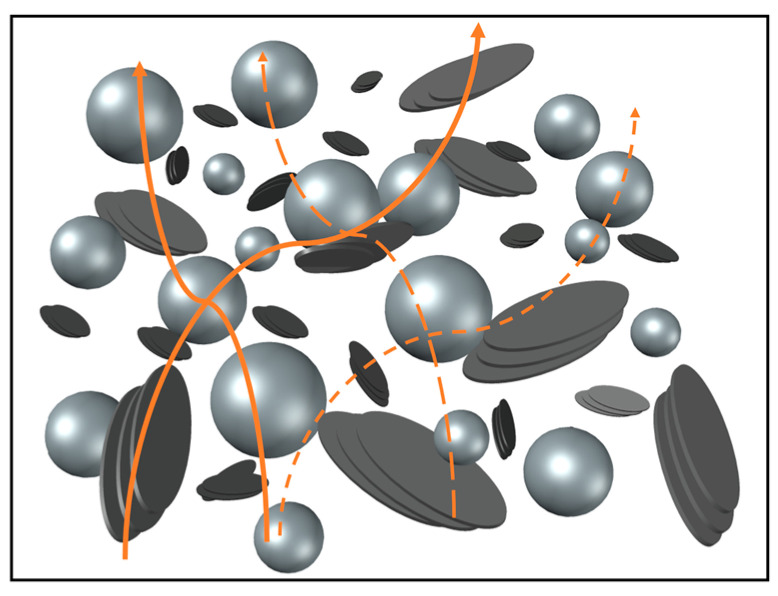
Schematic representation of the conductive paths (orange lines) in the HDPE hybrid filler system; the silver spheres represent the Al_2_O_3_ particles, while the different platelet particles represent the stacked layers of graphene; the big particles being EG and the small particles being G.

**Figure 7 polymers-15-00039-f007:**
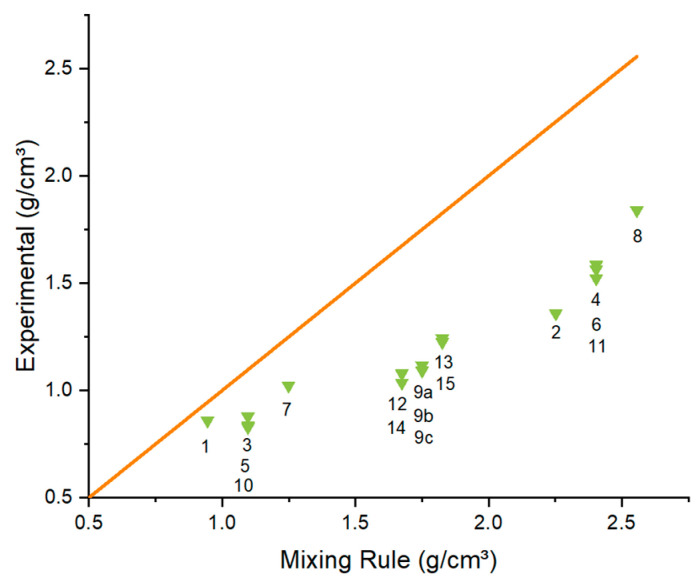
Densities of the compression moulded cubes as function of the densities according to the mixing rule (green) with the numbers on the graph representing the different samples (Table 1). The orange line indicates the perfect case that the mixing rule is valid.

**Figure 8 polymers-15-00039-f008:**
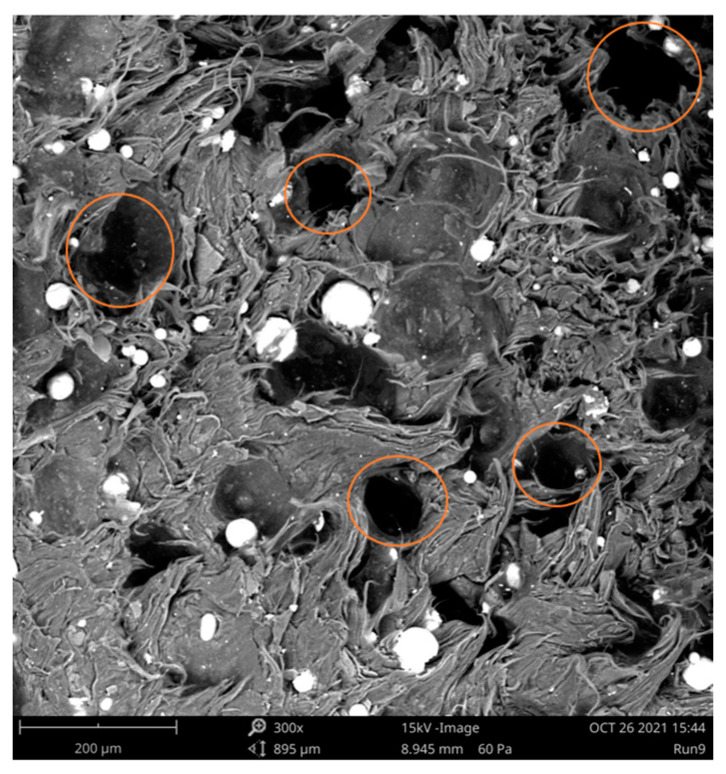
SEM-analysis of the air voids present in the compression moulded bricks of Sample 9, indicated in orange.

**Figure 9 polymers-15-00039-f009:**
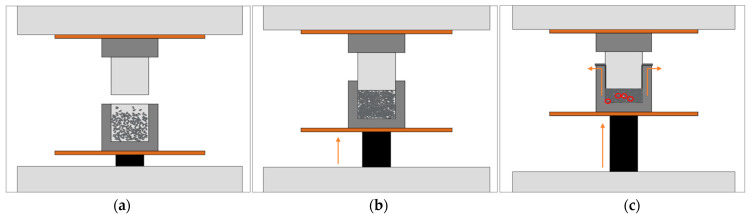
Compression moulding. (**a**) Pellets inside the mould. (**b**) Step 1: Melting the material for 10 min without pressure. (**c**) Step 2: Compression moulding with pressure until material flows out of the mould.

**Table 1 polymers-15-00039-t001:** Design matrix of the FCCD.

Sample/Run	Coordinates	Pattern	Al_2_O_3_	Graphite (G)	Exp. Graphite (EG)
1	(−1, −1, −1)	−−−	0	0	0
2	(+1, −1, −1)	+−−	45	0	0
3	(−1, +1, −1)	−+−	0	12.5	0
4	(+1, +1, −1)	++−	45	12.5	0
5	(−1, −1, +1)	−−+	0	0	12.5
6	(+1, −1, +1)	+−+	45	0	12.5
7	(−1, +1, +1)	−++	0	12.5	12.5
8	(+1, +1, +1)	+++	45	12.5	12.5
9a—central point	(0, 0, 0)	000	22.5	6.25	6.25
9b—central point	(0, 0, 0)	000	22.5	6.25	6.25
9c—central point	(0, 0, 0)	000	22.5	6.25	6.25
10	(−1, 0, 0)	−00	0	6.25	6.25
11	(+1, 0, 0)	+00	45	6.25	6.25
12	(0, −1, 0)	0−0	22.5	0	6.25
13	(0, +1, 0)	0+0	22.5	12.5	6.25
14	(0, 0, −1)	00−	22.5	6.25	0
15	(0, 0, +1)	00+	22.5	6.25	12.5

**Table 2 polymers-15-00039-t002:** Thermal conductivity results; relative data as normalized vs. reference values.

Sample/Run	Pattern	Filler Content (rel. m.%)			Normalized Thermal Conductivity (−)
		Al_2_O_3_	Graphite (G)	Exp. Graphite (EG)	
1	−−−	0	0	0	0.33
2	+−−	2	0	0	0.53
3	−+−	0	2	0	0.60
4	++−	2	2	0	1.44
5	−−+	0	0	2	0.74
6	+−+	2	0	2	1.72
7	−++	0	2	2	1.31
8	+++	2	2	2	3.23
9a—central point	000	1	1	1	0.95
9b—central point	000	*X*_1, ref_ = 1	*X*_2, ref_ = 1	*X*_3, ref_ = 1	*k*_ref_ = 1
9c—central point	000	1	1	1	0.98
10	−00	0	1	1	0.64
11	+00	2	1	1	1.57
12	0−0	1	0	1	0.64
13	0+0	1	2	1	1.35
14	00−	1	1	0	0.57
15	00+	1	1	2	1.51

**Table 3 polymers-15-00039-t003:** Extra compositions as validation of the response surface model (RSM) and the expected thermal conductivity of the extra compositions, in combination with the measured values.

Composition	V1	V2	V3
HDPE matrix (m.%)	69	69	59
CNT (m.%)	1	1	1
Al2O3 (m.%)	10	20	30
KS44 (m.%)	10	5	7.5
GFG75 (m.%)	10	5	7.5
Expected TC (W.m^−1^.K^−1^)	1.59	1.07	1.94
Measured TC (W.m^−1^.K^−1^)	1.53 ± 0.03	1.00 ± 0.03	1.81 ± 0.08

**Table 4 polymers-15-00039-t004:** Melt viscosity results at a shear rate of 3 s^−1^; relative data as normalized vs. reference values.

Sample/Run	Pattern	Filler Content (m.%; rel.)			$\mathrm{Relative}\;\mathrm{Viscosity}\;@\;\dot{\gamma }=3\;s^{-1}\;(-)$
		Al_2_O_3_	Graphite (G)	Exp. Graphite (EG)	
1	−−−	0	0	0	0.66
2	+−−	2	0	0	0.91
3	−+−	0	2	0	0.67
4	++−	2	2	0	1.09
5	−−+	0	0	2	0.81
6	+−+	2	0	2	1.42
7	−++	0	2	2	0.82
8	+++	2	2	2	1.76
9a—central point	000	1	1	1	1.01
9b—central point	000	*X*_1, ref_ = 1	*X*_2, ref_ = 1	*X*_3, ref_ = 1	*η*_ref_ = 1
9c—central point	000	1	1	1	0.94
10	−00	0	1	1	0.67
11	+00	2	1	1	1.27
12	0−0	1	0	1	0.78
13	0+0	1	2	1	0.94
14	00−	1	1	0	0.73
15	00+	1	1	2	1.21

**Table 5 polymers-15-00039-t005:** Density results of the prepared composites in combination with the density according to the mixing rule.

Sample/Run	Experimental Density (g/cm^3^)	Mixing Rule Density (g/cm^3^)
1	0.86	0.94
2	1.36	2.25
3	0.84	1.10
4	1.52	2.40
5	0.83	1.09
6	1.56	2.40
7	1.02	1.25
8	1.84	2.56
9a—central point	1.12	1.75
9b—central point	1.12	1.75
9c—central point	1.09	1.75
10	0.88	1.10
11	1.58	2.40
12	1.08	1.67
13	1.22	1.83
14	1.04	1.67
15	1.24	1.83

## Data Availability

The authors confirm that the data supporting the findings of this study are available within the article.

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
