# Peer review of "Response-Surface-Methodology-Based Increasing of the Isotropic Thermal Conductivity of Polyethylene Composites Containing Multiple Fillers"

_polymers, 2022, doi:10.3390/polym15010039_

Round 1
Reviewer 2 Report
Personally, this manuscript can be accepted after the author has addressed the following issues.
1. Why did the author choose graphite and expanded graphite as thermal conductive fillers? Both of them are platelet-shaped fillers. The author should provide the dimensions of graphite and expanded graphite.
2. Compression methods are used in the preparation of samples. In fact, flaky thermal conductive packing can also be oriented under pressure induction. Authors should provide convincing data to demonstrate the isotropic thermal conductivity of the prepared composites (In-plane thermal conductivity and out-of-plane thermal conductivity).
3. Whether the model works for the composite without CNTs.
4. Figure 6 shows the conduction pathway. Why not take carbon nanotubes into account?
Author Response
Review 2:
- Why did the author choose graphite and expanded graphite as thermal conductive fillers? Both of them are platelet-shaped fillers. The author should provide the dimensions of graphite and expanded graphite.
When only spherical fillers are to be used, the filler loading needs to be high to obtain a high thermal conductivity. For this reason, this research combines spherical fillers with platelet-shaped fillers to improve the conductive path. In addition, G and EG are chosen for their commercial relevance as these fillers are relatively cheap and effective. Graphite is cheaper than Expanded graphite, and therefore, it was investigated if there would be significant difference between them. From the regression equation, it can be clearly stated that expanded graphite is better than graphite. We have clarified this as well in the text now and we have added the dimensions of the graphites. Thank you for your comment on this.
- Compression methods are used in the preparation of samples. In fact, flaky thermal conductive packing can also be oriented under pressure induction. Authors should provide convincing data to demonstrate the isotropic thermal conductivity of the prepared composites (In-plane thermal conductivity and out-of-plane thermal conductivity).
We appreciate your suggestion. From the SEM-images in Figure 5 and in the images of the other composites in the Supporting Information, it can be seen that there is a good dispersion of all fillers and that the G and EG flakes are randomly oriented. These images therefore sufficiently show that the prepared composites have isotropic thermal properties. We have made this also more clear in the manuscript.
- Whether the model works for the composite without CNTs.
Thank you for you remark. However, in this particular research, the whole regression equation is based on a polymer matrix with a fixed amount of 1 m% CNTs. These CNTs are added since it has been showed in previous research that the addition of CNTs can have a significant synergistic improvement in the TC. (We have also this mentioned in the introduction in line 57-61) Since this research is performed with CNTs, it is likely not a model that works for the composite without CNTs. The amount of CNTs was not varied and investigated as an extra factor so the equation would not be valid for the polymer matrix without 1 m% CNTs.
- Figure 6 shows the conduction pathway. Why not take carbon nanotubes into account?
We already mentioned this in the manuscript in line 301-305: ‘The CNTs are very small compared to the other fillers (1.5 μm in length and nanometer-scale in cross-section in contrast to spherical Al2O3 fillers with a mean diameter of 41.2 μm) so that they are invisible in the SEM images.’
Reviewer 3 Report
The article contains interesting research on polymer modification. The authors presented the research results in an interesting way and reviewed the literature related to the topic.
However, the editorial and description of the analysis should be improved.
The abstract should state what results have been obtained.
Research conclusions should not be included in the abstract.
The symbols in the formulas need to be corrected
The symbol "*" is not allowed (equation 1, 2, 3, 4)
Fig 2b is not a figure but a table
Commas should be replaced with dots (description of the axes in the charts and in the text) – Fig 3, Fig 7
Figure 7 is not understandable (separate points), maybe it should be presented in a different way.
The method of citing literature in the text needs improvement in accordance with the journal's guidelines
Please specify whether the scheme (Fig. 1, Fig. 6, Fig. 9) was made by the authors of the article, and is not taken from the literature?
The article should be extended with an analysis of research results. The reasons for the registered changes should be explained (this is not exactly shown).
The article can be published after correction in accordance with the comments.
Author Response
- The abstract should state what results have been obtained.
Thank you for your suggestion. We modified the abstract in that case that we gave more information about the compositions that are investigated in this research, line 18-21: To optimize the thermal conductivity of high-density polyethylene, 15 hybrid filler composites containing aluminum oxide (up to 45 m%), graphite (up to 12.5 m%), expanded graphite (up to 12.5 m%), and carbon nanotubes (1 m%), have been designed, using response surface methodology (RSM) taking into account density and melt viscosity variations
- Research conclusions should not be included in the abstract.
We are very puzzled by this statement. For us and many papers in the field including Polymers it is a common procedure to do this. Hence, we have not accounted for this comment. The abstract is a summary of the whole work, including the results and discussion.
- The symbols in the formulas need to be corrected. The symbol "*" is not allowed (equation 1, 2, 3, 4)
We have modified these equations and have used an ‘x’ instead of the asterix for the multiplications.
- Fig 2b is not a figure but a table
Thank you for your remark. We have changed Fig 2b into Table 3.
- Commas should be replaced with dots (description of the axes in the charts and in the text) – Fig 3, Fig 7
We have modified this in the axes and description. Thank you for the suggestion.
- Figure 7 is not understandable (separate points), maybe it should be presented in a different way.
Thank you for your suggestion. We have added an extra table with the measured density results of the composites and the calculated results according to the mixing rule to make it more clear. In addition, instead of the separate points when the mixture rule is valid, we put a line.
- The method of citing literature in the text needs improvement in accordance with the journal's guidelines
We appreciate the suggestion and have checked all references and added the DOI everywhere.
- Please specify whether the scheme (Fig. 1, Fig. 6, Fig. 9) was made by the authors of the article, and is not taken from the literature?
Thank you for this remark. All figures were made by the authors themselves. None of these were taken from literature.
- The article should be extended with an analysis of research results. The reasons for the registered changes should be explained (this is not exactly shown).
The results and discussion of the results are shown together. We have highlighted more clearly what parts are analysis in the manuscript. In addition, for some parts, more extensive analysis of the research results has been written:
- Analysis of the dispersion of the SEM-images: line 286-293: ‘When the samples were cryogenically fractured, the Al2O3 particles did not break into two halves, but stayed intact leaving indentations in the other half of the sample. These SEM-images, in combination with the SEM-images of the other specimens, are also shown in the Supporting Information. From the SEM-images, it can thus be concluded that all fillers are well-dispersed and randomly oriented so that the prepared composites are assumed having isotropic thermal properties.’
- On the analysis of the main effects of the viscosity regression equation: line 333-346: ‘The coefficient terms of both Al2O3 and EG are much higher than the coefficient of G. In addition, the interaction terms and in Equation (7) are considerably high, which can be explained by Al2O3 particles preventing platelet graphite particles from orienting in the direction of the flow, giving more resistance to the polymer melt which can result in a viscosity increase. To determine the dominant filler for the increase in the composite viscosity, the viscosity can be predicted using the regression equation (7) and equation (5) to calculate the mass percentage into coded level. When the viscosity is calculated using 10 m% Al2O3 and no other fillers, and this viscosity is compared to the viscosity when only 10 m% G is used and when only 10 m% EG is used, significant differences in the viscosity at s-1 can be obtained. An addition of 10 m% Al2O3 results in a viscosity of 8337 Pa.s, whereas an addition of 10 m% EG leads to a viscosity of 9568 Pa.s. An addition of 10 m% G results in the lowest viscosity of 7820 Pa.s. From this, it can thus be stated that EG is the dominant filler for the viscosity increase. The table with these results are added in the Supporting Information.’
- Analysis and calculation of the density results: line 355-363: ‘Table 5 holds the density results from both the experimental samples and the calculated results according to the mixing rule.’ (We have added an extra Table) ‘In order to analyse a possible trend between the experimental density results and the density according to the mixture rule, Figure 7 shows the density measurements of the prepared composites as function of the density according to the mixing rule.’ Further analysis of these density results were already mentioned in line 368-376. The reason for this decrease in experimental density, being the increasing viscosity, was also mentioned in detail from line 377-400
Round 2
Reviewer 1 Report
After the revision, I suggest accepting this manuscript